# Impairment of Nutritional Status and Quality of Life Following Minimal-Invasive Esophagectomy—A Prospective Cohort Analysis

**DOI:** 10.3390/cancers16020266

**Published:** 2024-01-08

**Authors:** Grace Oberhoff, Lena Schooren, Florian Vondran, Andreas Kroh, Alexander Koch, Jan Bednarsch, Ulf P. Neumann, Sophia M. Schmitz, Patrick H. Alizai

**Affiliations:** 1Uniklinik Aachen, General-, Visceral- and Transplant Surgery, Pauwelsstr. 30, 52074 Aachen, Germany; grace.oberhoff@rwth-aachen.de (G.O.); fvondran@ukaachen.de (F.V.); akroh@ukaachen.de (A.K.); p.alizai@gk-bonn.de (P.H.A.); 2Uniklinik Aachen, Gastroenterology, Metabolic Diseases and Internal Intensive Care Medicine, Pauwelsstr. 30, 52074 Aachen, Germany; akoch@ukaachen.de; 3Uniklinik Essen, General-, Visceral- and Transplant Surgery, Hufelandstr. 55, 45147 Essen, Germany; jan.bednarsch@uk-essen.de (J.B.); ulf.neumann@uk-essen.de (U.P.N.); 4Gemeinschaftskrankenhaus Bonn, General- and Visceral Surgery, Prinz-Albert-Str. 40, 53113 Bonn, Germany

**Keywords:** esophageal cancer, minimal-invasive esophagectomy, malnutrition, quality of life, nutritional risk score

## Abstract

**Simple Summary:**

After minimal-invasive resection esophagectomy for cancer, patients experience significant impairments to nutritional supply and quality of life. The meticulous monitoring of vitamin status and potentially treatable nutritional intake is therefore mandatory.

**Abstract:**

Minimal-invasive resection of the esophagus for esophageal cancer has led to a relevant decrease in postoperative morbidity. Postoperatively, patients still suffer from surgical and adjuvant therapy-related symptoms impairing nutrition and quality of life. The aim of this study was to evaluate the nutritional status and associated symptoms six months after esophagectomy. Patients who attended follow-up examination six months after minimal-invasive esophagectomy were included. Blood and fecal tests, quality of life surveys (QLQ-C30 and QLQ-OG25) and nutritional risk screening (NRS) were performed. Twenty-four patients participated. The mean weight loss was 11 kg. A significant decrease in vitamin B12 (737 to 467 pg/mL; *p* = 0.033), ferritin (302 to 126 ng/mL; *p* = 0.012) and haptoglobin (227 to 152 mg/dL; *p* = 0.025) was found. In total, 47% of the patients had an impaired pancreatic function (fecal elastase < 500 µg/g). Physical (72 to 58; *p* = 0.034) and social functioning (67 to 40; *p* = 0.022) was significantly diminished, while self-reported global health status remained stable (52 to 54). The number of patients screened and found to be in need of nutritional support according to NRS score decreased slightly (59% to 52%). After MIE, patients should be meticulously monitored for nutritional status after surgery.

## 1. Introduction

Cancer of the esophagus is the eighth most common malignancy and sixth most common cancer-related death worldwide [1]. In recent years, survival rates of esophageal cancer have been increasing [1,2]. Therefore, quality of life (QoL) and the treatment of cancer-specific symptoms and malnutrition have shifted into the focus of therapy.

Esophageal cancer is often asymptomatic in the early stages of the disease, leading to delayed diagnosis [2,3]. By the time of diagnosis, dysphagia, dyspepsia, nausea, recurring vomiting or a lack of appetite are among the occurring symptoms [3,4,5]. These can all lead to a reduced food intake, resulting in weight loss, sarcopenia and nutritional deficiencies [6,7]. Some studies have shown that the preoperative nutritional status affects the postoperative outcome and prognosis [8]. Malnutrition and sarcopenia prior to surgery have been identified as risk factors for postoperative complications and negatively affect long-term survival [8,9]. Albumin, transferrin and prealbumin are established biochemical markers depicting the protein status [10,11]. Consequently, they can indicate protein deficiency in malnourished patients.

Patients with cancer of the upper GI tract are not only at risk of developing nutritional deficiencies and weight loss prior to surgery, but also after surgical treatment [12,13,14,15]. Underlying reasons for this are malabsorption, malnutrition and reduced food intake [13].

Meals should be high in calories and proteins, while portions should be smaller and should be eaten more frequently after upper GI surgery.

This requires a lot of effort and organization, and it is not uncommon that patients struggle with such drastic changes to their previous eating habits. Micronutrient deficiencies and GI symptoms following upper GI surgery may remain present even years and decades after upper GI surgery [16,17,18]. As malnutrition has been found to be related to depression in cancer patients, impaired QoL may be both the reason and result of restricted food intake and malnutrition [19].

Anemia following esophagectomy is common and anemia-related symptoms like fatigue or dyspnea can impair the patient’s quality of life [12,20]. Reasons for postoperative anemia in upper GI cancer patients are diverse. On the one hand, depending on the extent of resection, a reduction in the intrinsic factor can contribute to a malabsorption of vitamin B12 [21]. On the other hand, the effects of malnutrition due to reduced food intake also become apparent in B12, iron or folic acid deficiencies [20,22,23,24,25]. As anemia after upper GI surgery can be anticipated, it is important to monitor hemoglobin levels as well as MHC, MCV, iron, ferritin, vitamin B12 and folic acid to assess underlying deficiencies at follow up examinations.

Haptoglobin and C-reactive protein (CRP) belong to the acute phase proteins and increased serum concentrations can be found in inflammation, injury, infections and various malignant diseases including esophageal cancer [26,27,28,29,30,31,32]. Interleukin 6 (IL-6) is a proinflammatory cytokine and has been found to regulate growth in malignant tumor cells [33]. There also seems to be a correlation between high IL-6 levels and poor clinical features like heavy weight loss, advanced tumor stage and the presence of metastases as well as poor responses to treatment and prognosis [29,34,35]. The measurement of these parameters might allow for an estimation of the patients’ general inflammatory level, both before and after upper GI surgery.

Another clinically important nutritional impairment is pancreatic insufficiency following esophageal resection [12,13,36]. Exocrine pancreatic insufficiency is associated with maldigestion and malnutrition and contributes to impaired uptake of the fat-soluble vitamins A, D, E and K [37,38]. Further symptoms are diarrhea, steatorrhea, flatulence and unintentional weight loss [37]. Testing stool for pancreatic elastase-1 has been established as a marker for exocrine function of the pancreas and might therefore be a valuable screening parameter in patients following esophageal resection [37,39].

The aim of this study was to evaluate the short-term clinical outcome of patients undergoing minimal-invasive esophagectomy (MIE) for esophageal cancer and address possible therapeutic targets regarding malnutrition and quality of life.

## 2. Materials and Methods

### 2.1. Study Design

The study was conducted at the department of general, visceral and transplant surgery of the university hospital Aachen. Patients undergoing an esophagectomy in our center between December 2020 and October 2022 who agreed to participate in the study and attended a follow-up examination 6 months after surgery were included in this study. Patients that were operated on due to non-oncological reasons (fistulas, perforations and chemical burn) and patients that presented in an emergency setting were excluded. Further oncological follow-up examinations after 6 months were scheduled with the treating oncologist and the analysis was not part of this study. The study was reviewed and approved by our Ethics Committee (EK419/20). Written informed consent was obtained from all patients included in the study.

Demographic as well as clinical data were collected. The data used were either requested in the context of the medical history, determined by questionnaires, or originated from the internally used hospital information system (CGM medico, CompuGroup Medical SE & Co. KGaA, Koblenz, Germany). The data collection did not interfere with the treatment. The patients were treated according to the German guidelines, and the therapy concept was discussed and decided on by a weekly interdisciplinary tumor conference [40].

To evaluate the clinical outcome, we performed nutritional risk screenings (NRS), assessed weight as well as the body mass index, ran blood tests and asked the patients to fill out quality of life questionnaires, all prior to surgery and at the follow-up examination. Endoscopy and computer tomography were performed preoperatively and for follow-up examination.

### 2.2. Blood Tests

Blood tests included hemoglobin, hematocrit, leucocytes, CRP, ferritin, transferrin, haptoglobin, albumin, prealbumin, the vitamins A, B12 and D, IL-6 and phosphate. For standard values, see Table 1.

### 2.3. Fecal Samples

Patients were asked to bring a stool sample to the follow-up examination. The sample was supposed to be less than 24 h old and stored in a cool place before being handed to the staff. The samples were then tested for fecal elastase-1 as well as calprotectin by the clinic’s laboratory. For standard values, see Table 1.

### 2.4. Quality of Life Questionnaires QLQ-C30 and QLQ-OG25

To assess the patient’s subjective quality of life, we handed out health-related quality of life (HrQoL) surveys from the European Organization for Research and Treatment of Cancer (EORTC) QLQ-C30 (version 3) and QLQ-OG25, each in their German version [41]. While the QLQ-C30 questionnaire contains 30 general questions regarding HrQoL as well as more cancer-specific symptoms in oncological patients, the QLQ-OG25 queries more specific gastrointestinal cancer-related symptoms.

In 53 of the total 55 questions, the patients were asked to rank how heavily different symptoms and problems affected them on a scale from 1, “not at all”, to 4, “very much”. In the two remaining questions, patients were supposed to rate their overall health status and quality of life on a scale from 1, “extremely low”, to 7, “extremely high”. The responses were converted to scales from one to one hundred using linear transformation. Items of the QLQ-C30 were combined into six functional and nine symptom scales. The items in the QLQ-OG25 questionnaire were combined into 16 symptom scales. The higher the scores on the functional scales, the better the corresponding function, while higher scores on the symptom scales indicate greater impairment by a symptom or in a function.

### 2.5. NRS 2002

To quantify the patient’s risk of malnutrition, the Nutritional Risk Screening questionnaire (NRS 2002) recommended by the ESPEN guidelines was used [42]. A pre-screening made up of four questions should detect patients at risk for developing nutritional impairment. The main screening then evaluates the need for nutritional support. A score of three or more indicates the need for a nutrition plan. If the score is below three then screening should be repeated weekly.

### 2.6. Surgical Procedure

A minimally invasive subtotal esophagectomy was performed for esophageal cancer and esophagogastric junction adenocarcinoma type I and II. The procedure was carried out laparoscopically and thoracoscopically with a 2-field lymphadenectomy.

### 2.7. Statistical Analysis

Microsoft Excel (Microsoft Office. Microsoft Released 2023. Version 16.75) as well as IBM Statistical Package for Social Science (IBM SPSS Statistics for Macintosh, Version 28.0. Armonk, NY, USA) were used for all statistical analyses.

Data are indicated as mean (SD) unless stated otherwise. Data prior to surgery and at follow-up were compared using the paired *t*-Test. A two-sided *p* of <0.05 was considered statistically significant.

## 3. Results

In total, 24 patients with esophageal cancer who underwent MIE between December 2020 and October 2022 in our center agreed to participate in the study and attended the follow-up examination. In total, there were 51 oncological patients operated on in this period, and the remaining patients were either lost to follow up, died in the time to follow up or declined to participate (see Figure 1). The median follow up time was 6.4 months (±1.6). In our cohort, 83% of patients were male.

Almost all patients underwent neoadjuvant treatment (96%), mostly in the form of chemotherapy. Fifteen patients received chemo according to the FLOT regime, seven patients underwent radiochemotherapy (six of them followed the CROSS protocol consisting of paclitaxel and oxaliplatin and one patient received a combination of 5-FU and oxaliplatin in addition to radiation therapy) and one patient had radiation only. More than two thirds of the patients received adjuvant treatment: fifteen patients had adjuvant chemotherapy, and two patients received nivolumab.

Nineteen patients (79%) had no or minor complications (Clavien Dindo ≤ 3a), and five patients (21%) experienced major complications (Clavien Dindo ≥ 3b). There was no relapse of cancer in any of our patients at the time of follow-up, as examined by endoscopy and computer tomography. None of the patients had anastomotic stenosis or anastomotic insufficiency at time of follow-up. For more details on the patient cohort see Table 2.

### 3.1. Body Weight, BMI, Nutritional Risk Screening

In patients after MIE, weight and BMI were significantly lower 6 months after the operation compared to the values prior to operation: weight 82.9 kg versus 72.0 kg (*p*-value < 0.001); BMI 27.5 kg/m^2^ versus 23.9 kg/m^2^ (*p*-value < 0.001). Mean weight loss was 12.9%. Fourteen patients (58%) lost more than 10% of their preoperative body weight.

At the first visit, approximately half of our patient cohort (59%) had an NRS of 3 or more, indicating the need for nutritional support. Then, 6 months after esophagectomy with 52% requiring nutritional support, the amount stayed approximately the same (see Table 2 and Figure 2 and Figure 3).

### 3.2. Laboratory Tests

There were no statistical differences in the values for leucocytes, hemoglobin, hematocrit, MCV, MCH, albumin, vitamin A, vitamin D, transferrin, IL-6, CRP, phosphate and prealbumin, while statistically significant differences could be seen for vitamin B12, ferritin and haptoglobin (vitamin B12 737.2 ± 471.9 versus 466.5 ± 178.8, *p*-value 0.033; ferritin 301.6 ± 279.7 versus 125.5 ± 118.0, *p*-value 0.012; haptoglobin 2271 ± 143.4 versus 152.3 ± 54.3, *p*-value 0.025). For a complete overview of the laboratory results, see Table 3.

### 3.3. Fecal Samples

Eight patients provided fecal samples at the first visit, and fifteen patients at the time of follow-up. Two patients reported taking pancreatic enzyme replacement therapy at the time of follow-up. Six months after upper GI surgery, mean fecal elastase-1 was 581.8 μg/g (±379.4). Reduced elastase-1 values below 500 µg/g were seen in seven patients (47%), of which one patient had <200 μg/g, indicating pancreatic insufficiency. Approximately half our cohort (53%) showed elevated calprotectin levels >50 μg/g at time of follow-up. For results of the fecal samples see Table 4.

### 3.4. Quality of Life

Six months after MIE, global health status remained stable compared to preoperative values. We found statistically significant decreases in physical functioning and social functioning after the operation (*p*-values 0.034 and 0.022, resp.). For an overview of QoL results see Table 5.

Concerning cancer-specific symptoms, there was a significant increase in the symptoms of pain, dyspnea, insomnia, eating with others and trouble with coughing. After MIE, patients reported significantly less constipation than prior to surgery (20.8 ± 29.5 versus 2.1 ± 19.1; *p*-value 0.007). For an overview of the assessed cancer-specific symptoms, please refer to Table 5.

## 4. Discussion

Tumors of the esophagus are challenging to treat and might have a considerable impact on the quality of life and nutritional status of patients even after curative treatment [13,43,44].

In this study, at 6-month follow-up, the mean body weight loss was 13%, respectively, 11 kg. In a multidisciplinary survivorship clinic, mean weight loss has been reported to be 8.5% six months after esophagectomy [45]. Heneghan et al. observed heavy weight loss (>10% of body weight) in 16% at follow-up 6 months after upper GI cancer surgery [13]. The percentage increased to 49% of the patients at 24 months postoperatively [13]. Heavy weight loss was detected in almost 60% of our patient cohort. Nevertheless, mean BMI was 24 kg/m^2^ at follow-up, which was within the normal range of a healthy BMI [46]. As the BMI does not reflect the body mass composition, patients might be affected by sarcopenia without being diagnosed in this study. Malnutrition and weight loss have enormous clinical impact and therefore detecting underlying conditions and deficiencies is of utmost importance.

Ferritin levels in this study drastically decreased postoperatively, which is in line with Janssen et al., who reported decreased iron and ferritin levels after MIE [12]. This could have different explanations: For one, ferritin is known to be an acute phase protein and has been found to be elevated in acute or chronic inflammation as well as in malignancies or other diseases [47]. Therefore, it could be discussed whether the ferritin levels were increased prior to surgery due to inflammation and acute malignancy. Nevertheless, it could also reflect decreasing iron storage, putting the patient at risk of developing microcytic anemia. Heneghan et al. found ferritin levels to be increased one month after surgery, while reaching lowest levels at 6-month follow-up before increasing again and finally showing no significant difference after 18 to 24 months compared to preoperative values [13].

A similar trend was seen in haptoglobin levels, also an acute phase protein [48]. Reasons for decreased haptoglobin levels may be hemolysis, allergic reactions or malnutrition [26,27]. CRP is an established marker for disease activity and elevated CRP levels seem to be associated with advanced tumor stage and poor prognosis [32,49,50,51,52]. The drop of haptoglobin in our cohort and no significant change in hemoglobin levels as well as decreased CRP and IL-6 levels at the time of follow-up support the first assumption.

The significant decrease in vitamin B12 levels 6 months after MIE we found was in line with reports by van Hagen et al. [53]. We found hemoglobin levels to be below the recommended values in two female and fourteen male patients prior to surgery as well as postoperatively. While reduced vitamin B12 and folic acid levels cause a macrocytic hyperchromic form of anemia, a lack of iron rather leads to microcytic hypochromic anemia [20,54]. As most patients showed MCH and MCV values within the normal range, this might indicate a lack of both iron and B12 as responsible for developing anemia. As the risk of developing anemia is already increased due to upper GI surgery, hemoglobin levels and possible deficiencies should be closely checked up on and, if necessary, supplementation should be initiated whenever indicated.

The average level of fecal elastase-1 dropped compared to preoperative values, but a level below 200 μg/g was only found in one patient, indicating a manifest exocrine pancreatic insufficiency. Other studies have also shown reduced fecal elastase-1 levels after esophagectomy [13,37]. As pancreatic insufficiency leads to an impaired uptake of fat-soluble vitamins and malabsorption, making it difficult for patients to maintain weight, it is important to monitor pancreatic function in the long term [55]. The substitution of pancreatic enzymes can restore weight in cases of pancreatic insufficiency, which should therefore be screened for [37,55].

Calprotectin levels after MIE were elevated in more than half of our patients. Reasons for this could be ongoing inflammation in the gastrointestinal tract or issues related to occurring digestive symptoms [56]. Other studies showed elevated calprotectin levels in gastrointestinal cancers, including esophago-gastric cancer [57,58,59]. To our knowledge, calprotectin levels after esophagectomy have not been assessed yet. Due to the small number of fecal samples, we did not calculate significances regarding elastase-1 and calprotectin levels.

Regarding QoL, the patients in our cohort were dealing with general symptoms like pain, dyspnea, coughing and insomnia 6 months after surgery, and on the other hand food-related symptoms such as trouble with eating, odynophagia and dysphagia remained relatively stable. As almost two thirds of our patients received adjuvant (chemo)therapy, it remains unclear whether the symptoms were side effects of the chemotherapy or related to the surgery. Our findings of improved global health status scores after MIE seem paradoxical at first, especially regarding impaired function in all scores. However, considering the time and context in which our data were collected, patients may have felt less anxious and mentally healthier having completed the aggressive treatment compared to the first QoL assessment, when surgery and chemotherapy in most cases were still lying ahead. A study from Sweden also found that anxiety and fear reduced with time as patients adjusted to postoperative changes after gastrectomy and esophagectomy [60]. A Dutch registry study reported the highest prevalence of gastrointestinal symptoms up to 3 months after esophagectomy or gastrectomy, while 9 to 12 months after the operation values were reported to lower again [16]. Boshier et al. described the presence of gastrointestinal symptoms as associated with overall reduced HrQoL, especially emotional and social function, in a multicenter setting [18]. Long-term HrQoL can be impaired more than two years after gastrectomy and esophagectomy [61]. Fuchs et al. also, consistent with our data, reported significant impairments for physical function, dyspnea and reflux after MIE [61]. While reflux after MIE did not reach statistical significance in our cohort, there could be a trend towards a higher symptom burden noted 6 months after MIE. Gastrointestinal symptoms and negatively affected quality of life have been reported to remain relatively constant one year after the operation. Contrary to our data, in most studies patients underwent an open approach, and data on long-term survivors of minimally invasive surgery is still rare [18].

There are some limitations to our study: The study cohort was relatively small. A reason for this was the limited capacities for follow-up appointments during the COVID-19 pandemic. Furthermore, as participation in a follow-up examination was voluntary, there were quite a few patients who were either not interested or not able to attend such an appointment. One could assume that patients who were highly symptomatic with low quality of life were particularly interested in a follow-up to consult with a physician again. On the other hand, there might have been patients who were too ill or still undergoing (oncologic) treatment and therefore did not come to the follow-up. In addition, some patients lived quite far away from our clinic and were in aftercare with their local oncologist or general physician. Furthermore, the preoperative data were collected at first contact with our clinic. Therefore, we met patients at different times in their treatment. Some had already completed neoadjuvant therapy, while it was still lying ahead for others, making it difficult to compare results. Collecting stool samples turned out to be more difficult than expected, as some patients forgot to take a sample, and as the stool was not supposed to be older than 24 h, some patients were not able to provide us with a sample as they were struggling with irregular digestion.

Our follow-up being 6 months after surgery met the time when gastrointestinal symptoms and weight loss have been reported to be the heaviest [45]. To track the development of the nutritional status, further follow-up examinations are the subject of future research.

However, this prospective cohort study gives important insights into nutritional deficiencies and quality of life impairments after curative minimal-invasive esophagectomy. These results should be taken into account for nutritional therapy and substitution after esophageal surgery.

## 5. Conclusions

Our study shows that patients face a multitude of challenges 6 months after curative intended MIE. For one, patients lose a considerable amount of body weight after surgery. Furthermore, blood levels of ferritin and vitamin B12 significantly decrease, which may cause anemia. Several conditions, such as pancreatic insufficiency and vitamin deficiency, are easily treatable and should therefore be screened for. Furthermore, patients seem to be afflicted by general symptoms like dyspnea, insomnia, pain, eating with others and trouble with coughing. These symptoms may be reasons that patients feel significantly impaired in their physical and social function. However, this does not seem to affect the overall perceived global health status.

As patients are at risk of developing several nutritional deficiencies, it is important to regularly check corresponding blood values, ask about specific gastrointestinal symptoms and evaluate their quality of life.

## Figures and Tables

**Figure 1 cancers-16-00266-f001:**
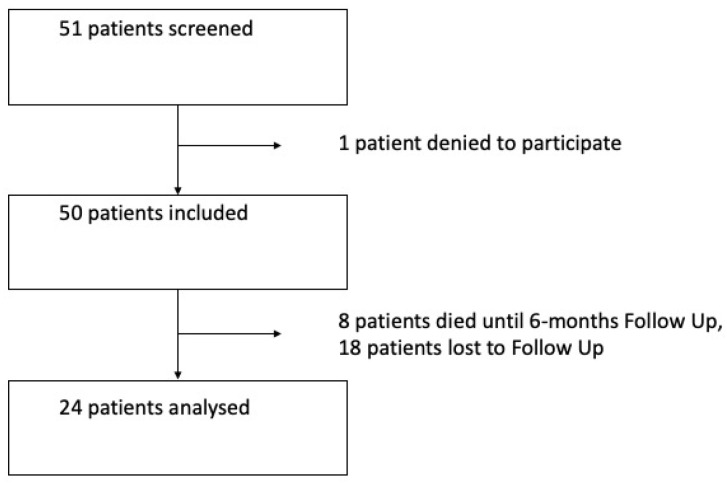
Flowchart of patients included in the study.

**Figure 2 cancers-16-00266-f002:**
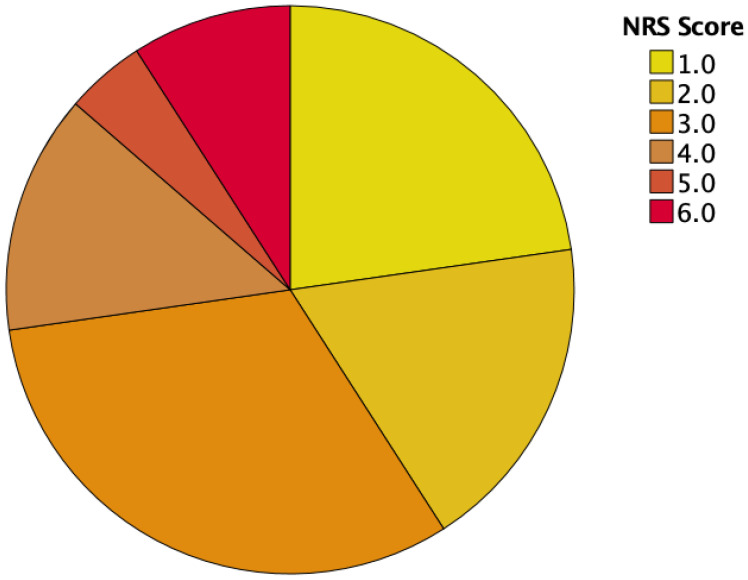
NRS scores prior to surgery.

**Figure 3 cancers-16-00266-f003:**
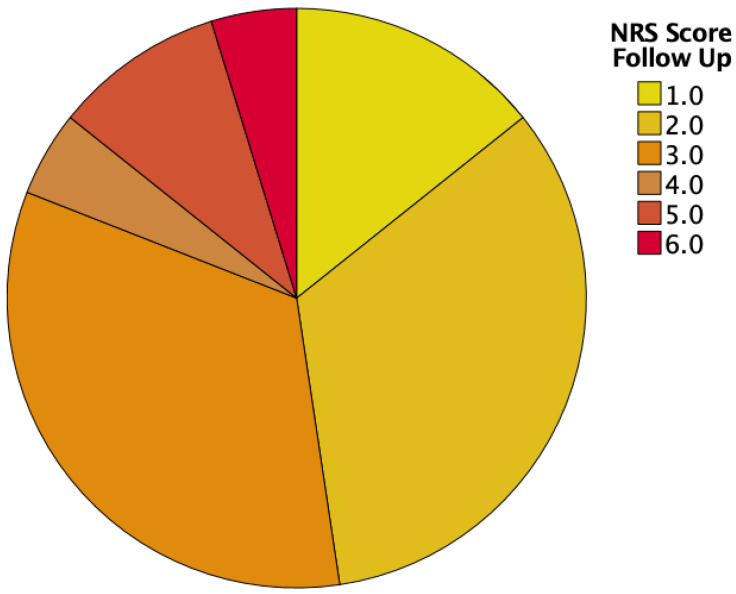
NRS scores at follow-up.

**Table 1 cancers-16-00266-t001:** Standard values according to our laboratory.

Parameter	Standard Value	
Albumin	3.5–5.2 g/dL	
Prealbumin	20–40 mg/dL	
Vitamin A	300–700 μg/L	
Vitamin B12	197–771 pg/mL	
Vitamin D3 (25-OH-Vitamin D)	Deficit < 20 ng/mL; toxic > 100 ng/mL	
CRP	<5 mg/L	
IL-6	<7 pg/mL	
Transferrin	200–360 mg/dL	
Ferritin	♀ 15–150 ng/mL	♂ 30–400 ng/mL
Haptoglobin	30–200 mg/dL	
Phosphate	0.81–1.45 mmol/L	
Leucocytes	♀ 4.0–10.0/nL	♂ 4.2–9.1/nL
Hemoglobin	♀ 11.2–15.7 g/dL	♂ 13.7–17.5 g/dL
Hematocrit	♀ 34.1–44.9%	♂ 40.1–50.0%
MCV	♀ 79.4–94.8 fl	♂ 79.0–92.2 fl
MCH	25.6–32.2 pg	
Elastase	>200 µg/g	
Calprotectin	<50 µg/g	

**Table 2 cancers-16-00266-t002:** Information on study population.

Patient Characteristics	All (*n* = 24)
Age	64.3 years (±8.6)
Male	20 (83%)
Arterial hypertension	14 (58%)
History of smoking	12 (5%)
Chronic obstructive pulmonary disease	2 (8%)
Coronary heart disease	6 (25%)
Diabetes mellitus type 2	7 (29%)
Obesity (BMI > 25 kg/m^2^)	6 (25%)
Neoadjuvant therapy	23 (96%)
- FLOT	15 (65%)
- RCT	7 (30%)
- Radiation	1 (4%)
Adjuvant therapy	17 (71%)
- Chemotherapy	15 (88%)
- Nivolumab	2 (12%)
ASA 2	6 (25%)
ASA 3	17 (71%)
ASA 4	1 (4%)
Clavien-Dindo ≤ 3a	19 (79%)
Clavien-Dindo ≥ 3b	5 (21%)
NRS < 3	9 (41%)
NRS ≥ 3	13 (59%)

**Table 3 cancers-16-00266-t003:** Blood values prior to and after surgery.

	Prior to Surgery	After Surgery	*p*-Value
**Weight**	**82.9 kg (14.6 kg)**	**72.0 kg (14.0 kg)**	**<0.001**
**BMI**	**27.5 kg/m^2^ (3.9 kg/m^2^)**	**23.9 kg/m^2^ (3.9 kg/m^2^)**	**<0.001**
Leucocytes	7.0/nL (2.7/nL)	5.9/nL (1.2/nL)	0.058
Hemoglobin	12.2 g/dL (2.2 g/dL)	12.5 g/dL (1.9 g/dL)	0.459
Hematocrit	37.0% (6.2%)	38.4% (6.1%)	0.267
MCV	87.3 fl (5.6 fl)	88.4 fl (6.8 fl)	0.125
MCH	28.7 pg (2.3 pg)	28.8 pg (2.7 pg)	0.145
Albumin	4.2 g/dL (0.3 g/dL)	4.3 g/dL (0.4 g/dL)	0.329
Vitamin A	595.5 μg/L (163.8 μg/L)	529.1 μg/L (142.6 μg/L)	0.284
Vitamin D	18.7 ng/nL (12.2 ng/mL)	21.0 ng/mL (10.9 ng/mL)	0.362
**Vitamin B12**	**737.3 pg/mL (471.9 pg/mL)**	**466.5 pg/mL (178.8 pg/mL)**	**0.033**
Transferrin	257.8 mg/dL (59.7 mg/dL)	265.2 mg/dL (60.5 mg/dL)	0.538
**Ferritin**	**301.6 ng/mL (279.7 ng/mL)**	**125.5 ng/mL (118.0 ng/mL)**	**0.012**
**Haptoglobin**	**227.1 mg/dL (143.4 mg/dL)**	**152.3 mg/dL (54.3 mg/dL)**	**0.025**
CRP	2.6 mg/L (1.6 mg/L)	1.4 mg/L (1.0 mg/L)	0.018
IL-6	7.5 pg/mL (6.3 pg/mL)	5.1 pg/mL (5.4 pg/mL)	0.353
Phosphate	1.1 mmol/L (0.2 mmol/L)	1.1 mmol/L (0.2 mmol/L)	0.518
Prealbumin	26.3 mg/dL (4.6 mg/dL)	24.5 mg/dL (5.5 mg/dL)	0.188

Values in mean ± standard deviation; statistical significances are highlighted in bold.

**Table 4 cancers-16-00266-t004:** Results of fecal samples.

	Prior to Surgery	After Surgery
Elastase-1	1692.3 μg/g (±664.0)	581.8 μg/g (±379.4)
Calprotectin	67.3 μg/g (±45.4)	54.6 μg/g (±40.9)

Values in mean ± standard deviation.

**Table 5 cancers-16-00266-t005:** QoL scores prior to and 6 months after surgery.

	Prior to Surgery	After Surgery	*p*-Value
Global Health Status	52.2 (20.3)	54.2 (21.2)	0.776
**Physical functioning**	**71.8 (19.7)**	**57.9 (27.2)**	**0.034**
Role functioning	53.3 (40.4)	40.0 (27.3)	0.238
Emotional functioning	61.1 (29.5)	56.1 (24.5)	0.591
Cognitive functioning	90.0 (18.7)	75.6 (28.8)	0.072
**Social functioning**	**66.7 (28.2)**	**40.0 (33.8)**	**0.022**
Fatigue	46.2 (26.6)	59.7 (25.6)	0.169
Nausea and vomiting	26.0 (27.9)	19.8 (28.0)	0.591
**Pain**	**19.8 (22.1)**	**43.2 (34.3)**	**0.041**
**Dyspnea**	**22.9 (29,1)**	**47.9 (29.7)**	**0.013**
**Insomnia**	**33.3 (32.2)**	**47.9 (421)**	**0.048**
Appetite loss	39.6 (37.0)	52.1 (38.4)	0.287
**Constipation**	**20.8 (29.5)**	**2.1 (19.1)**	**0.007**
Diarrhea,	37.8 (33.0)	45.6 (33.6)	0.235
Financial difficulties	35.7 (35.7)	42.9 (35.6)	0.533
Dysphagia	21.9 (29.1)	20.1 (21.9)	0.858
Eating	42.7 (25.6)	50.5 (26.1)	0.297
Reflux	13.5 (18.5)	30.2 (28.7)	0.076
Odynophagia,	19.8 (28.7)	19.8 (26.0)	1.000
Pain and discomfort	15.6 (30.1)	33.3 (30.4)	0.129
Anxiety	68.8 (28.5)	65.6 (28.8)	0.580
**Eating with others**	**6.3 (18.1)**	**25.0 (37.5)**	**0.045**
Dry mouth	35.6 (36.7)	24.4 (36.7)	0.334
Trouble with taste	52.1 (43.8)	39.6 (38.9)	0.383
Body image	28.9 (27.8)	40.0 (40.2)	0.353
Trouble swallowing saliva	16.7 (24.3)	16.7 (27.2)	1.000
Choked when swallowing	14.6 (17.1)	12.5 (20.6)	0.718
**Trouble with coughing**	**20.8 (16.7)**	**50.0 (34.4)**	**0.001**
Trouble talking	6.7 (13.8)	8.9 (26.6)	0.774
Weight loss	29.2 (34.2)	41.7 (39.4)	0.188
Hair loss	52.4 (46.6)	33.3 (43.0)	0.280

Scores indicated as mean ± standard deviation; statistical significances are highlighted in bold.

## Data Availability

Data are contained within the article.

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
