# Peer review of "Impairment of Nutritional Status and Quality of Life Following Minimal-Invasive Esophagectomy—A Prospective Cohort Analysis"

_cancers, 2024, doi:10.3390/cancers16020266_

Round 1
Reviewer 1 Report
Comments and Suggestions for Authors
Congratulations for your great study. I have some questions.
1. Could you define the relationships between the nutrition targets (ferritin…) and esophageal cancer?
2. How about other parameters, such as albumin level, BMI, body fat component?
Author Response
Dear Reviewer 1,
We kindly thank you for taking your time to read our manuscript and for your feedback. We would like to answer your questions. Due to the paper’s length we already tried to keep some background information short. Upon revision of our manuscript, we tried to emphasize the connection between the assessed parameters and esophageal cancer even more in the introduction.A short background on the role of ferritin is given in the discussion (lines 243-252). The other parameters that you asked about we tried to cover with the section on symptoms of esophageal cancer resulting in sarcopenia as they are all related (see lines 50-57). I hope this answers your questions to your content. We again thank you very much for your time and appreciation of our manuscript.
Reviewer 2 Report
Comments and Suggestions for Authors
I do not understand what the primary end point and what authors would like to. Unfortunately, I could not understand what the strength of this article was, as there was no comparison with conventional surgery. Also, although the short term is important, if you are going to take the prospective study, I would like you to consider whether the patient will recover in the long term and whether good results can be obtained with intervention.
Comments on the Quality of English LanguageEnglish is generally ok.
Author Response
Dear Reviewer 2,
We kindly thank you for taking your time to read our manuscript and for your feedback. We are sorry that the aim and idea of our study were not clear in the first place. We now revised the whole manuscript and focused more on the aim of our study. We made the connection between the assessed parameters and esophageal cancer clearer. Our aim was not to compare between conventional and minimally invasive surgery but to assess treatable deficiencies after esophageal surgery in minimally invasive performed esophagectomies. We emphasized this in the introduction and discussion. We agree that interventional studies are best to discover possible treatment pathways but we also believe there are important findings in our study that should be considered by physicians treating patients after esophagectomy. Based on our findings, an intervention, e.g. with nutritional substitution or substitution of pancreatic enzyme would be a great idea for further studies. We hope you find our revisions helpful and reconsider our manuscript for publication now.
Reviewer 3 Report
Comments and Suggestions for Authors
The study assessed 24 patients before and 6 months after esophagectomy in the nutritional quality and routine life. It is found that the minimal invasive operation affects the life of the patients in multiple ways. The findings are nothing out of ordinary.
A minor correction, in the abstract, it is said " The amount of patients screened to be in need of nutritional support according to NRS score de- 39 creased slightly (59% to 52%)." It is better to use the word "number" instead of "amount".
Comments on the Quality of English LanguageThe manuscript is written fairly well.
Author Response
Dear Reviewer 3,
We kindly thank you for taking your time to read our manuscript and for your feedback. We changed the sentence in the abstract according to your suggestion. We furthermore improved the introduction by shortening it and focusing more and added several parts to the discussion. We hope you find it suitable for publication now.
Reviewer 4 Report
Comments and Suggestions for Authors
This study reports on functional and nutritional parameters after minimally-invasive esophagectomy for esophageal cancer. The number of analyzed patients is rather small, which obviously limits the external validity of the data. Moreover, there is already a considerable amount of evidence on the topic from larger studies. Nonetheless, I think the results are important for clinicians as they add to the body of evidence and provide a good impression of postoperative issues in these patients.
I have a number of suggestions how the manuscript could be improved:
- The introduction could be shortened.
- You should present the number of patients operated on during the entire study period, the number of patients screened for study participation, the number of patients enrolled in the study and the number of patients lost to follow-up in a flowchart. The entire presentation of the study should follow the STROBE statement.
- Do you have any information if patients received postoperative substitution of pancreatic enzymes which you could present?
- I don't think the statement "Patients after MIE showed a slight trend towards an improved global health status 218 at follow-up compared to preoperatively" is justified. There is virtually no difference and the p-value is 0.776.
- The lack of QoL data prior to initiation of neoadjuvant treatment (i.e. at first diagnosis) is a limitation which needs to be addressed.
- Is there any data from the literature (I assume you don't have them at hand for your own patients) on how the parameters under study change after open as compared to minimally-invasive esophagectomy? These could well inform the discussion.
Author Response
Dear Reviewer 4,
We kindly thank you for taking your time to read our manuscript and for your feedback. We shortened the introduction and tried to focus more on the implications of patients after esophagectomy. We furthermore included the number of patients operated in the study period and the number of finally included participants. At time of follow up we asked all patients about their medication, two patients reported to receive pancreatic enzyme replacement therapy. We included this in the results (line 280). We furthermore changed the sentence regarding global health status to “Six months after MIE global health status remained stable compared to preoperative values.“. We absolutely agree that lack of QoL data prior to initialtion of neoadjuvant treatment is a limitation of this study and added this point according to your suggestion. This was due to patients’ schedules that often did not include presenting at the surgincal center prior to initiation of neoadjuvant therapy. Concerning your last question: From what we know there is no data on nutritional deficiencies after minimally-invasive esophagectomy compared to open esophagectomy. As we performed the operations only minimally-invasively we cannot give any information about this from our own patient cohort. We would like to thank you again for your time in reading our manuscript and hope you find our revisions helpful.
Round 2
Reviewer 4 Report
Comments and Suggestions for Authors
Thank you for having revised the manuscript according to some of my suggestions. I have no further comments.